# *Leptospira interrogans* Serovar Icterohaemorrhagiae Failed to Establish Distinct Infection in Naïve Gilts: Lessons Learned from a Preliminary Experimental Challenge

**DOI:** 10.3390/pathogens12010135

**Published:** 2023-01-13

**Authors:** Romana Steinparzer, Sophie Duerlinger, Friedrich Schmoll, Adi Steinrigl, Zoltán Bagó, Denise Willixhofer, Osaid Al Salem, Sarolta Takács, Christian Knecht, René Renzhammer, Ilse Schwendenwein, Andrea Ladinig, Christine Unterweger

**Affiliations:** 1Institute for Veterinary Disease Control, Austrian Agency for Health and Food Safety (AGES), Robert Koch Gasse 17, 2340 Moedling, Austria; 2University Clinic for Swine, Department for Farm Animals and Veterinary Public Health, University of Veterinary Medicine, 1210 Vienna, Austria; 3Clinical Pathology Platform, University of Veterinary Medicine, 1210 Vienna, Austria

**Keywords:** Icterohaemorrhagiae, gilts, experimental infection, urogenital tract, vaginal swabs, cultivation, qPCR, MAT, seroconversion, tubulo-interstitial nephritis

## Abstract

*Leptospira* is a pathogen involved in fertility problems in pigs. Nevertheless, little information is available on pathogenicity, transmission, tissue tropism, and immune response. The objective of this preliminary study was to induce a diagnostically detectable infection in naïve gilts using *Leptospira interrogans* serovar Icterohaemorrhagiae to gain the knowledge required for designing a large-scale trial. Eight seronegative fertile gilts were divided into three groups: control (n = 2), challenge (n = 3; 10 mL of 10^8^ leptospires/mL intravenously), and contact (n = 3). A daily clinical examination and periodic sampling of blood, urine, and vaginal swabs were performed until four weeks after infection when necropsy was undertaken. Seroconversion of infected animals was detected first by a microscopic agglutination test (MAT) between four and seven days after inoculation. No clinical signs were observed except pyrexia. Laboratory data primarily remained within reference intervals. *Leptospira* were undetectable in all groups by real-time PCR (sera, urine, vaginal swabs, and tissue samples) and bacterial culture (urine and tissue samples). However, histologic evidence for tubulo-interstitial nephritis could be found. Based on the study results and limitations, questions to be solved and approaches to be reconsidered are raised for the conduction of further experimental studies to understand the pathogenesis and the role of Icterohaemorrhagiae in pig health.

## 1. Introduction

Leptospirosis is a bacterial, zoonotic disease caused by pathogenic *Leptospira* species that affects humans and several mammalian species. *Leptospira* are transmitted through direct contact or indirectly through the environment such as contaminated water or manure [1] and penetrate defective skin or mucosa [2]. Altogether, 66 *Leptospira* species, with more than 30 serogroups and 300 serovars, are currently known [3,4]. *Leptospira* spp. have been associated with reproductive disorders in pigs for decades. Strains belonging to the serogroups Australis and serovar Pomona were described as being well adapted to swine, which are considered maintenance hosts, whereas serovars of the Icterohaemorrhagiae, Grippotyphosa and Tarassovi serogroups incidentally occur in pigs [5]. The occurrence of porcine leptospiral infections was confirmed worldwide, mainly by antibody detection [6,7]. Leptospiral antibodies against serovar Icterohaemorrhagiae, which belongs to the serogroup Icterohaemorrhagiae and is classified genetically in subclade P1 [4], are frequently detected [7,8,9]. During the last decade, 19.8% of the detected *Leptospira* antibodies in sows from German swine stocks were directed against Icterohaemorrhagiae [8]. In Austria, this number reached about 50% in 2021 (unpublished data). Those numbers are of particular interest since this serovar has never been identified as the causative agent of fertility problems in sows by examining organs or urine using direct detection methods within previous decades. This can be explained by two possible reasons: technical laboratory diagnostic limitations and knowledge gaps in the epidemiology and pathogenesis of this serovar, followed by inappropriate diagnostics to detect *Leptospira* strains. Technically, direct molecular diagnostic methods are capable of detecting *Leptospira* on a species level [10], but not on a serovar level [11]. One would need isolation and further sequencing to identify a *Leptospira* serovar. However, leptospiral isolation is hardly ever performed in routine diagnostics, because of the special growth conditions and the rather excessive length of the cultivation period [12]. For routine diagnostics, a combination of PCR and serological tests is frequently applied. Knowledge gaps in epidemiology and pathogenesis in pigs could be closed by experimental studies. It would be essential to define the optimal sample matrices for direct detection, to clarify the optimal time point of sampling, and to understand the transmission of the pathogen. Current knowledge mainly derives from experimental infection trials in rodents [13,14,15,16,17]. Getting full information for a host animal from a model using different species is not entirely possible because there can be considerable differences between species and even between animal model strains [18]. Some leptospiral experimental studies were conducted in pigs, mainly with the serovars Pomona and Hardjo [19,20]. Of note, the pathogenesis of the disease can vary depending on the serovar [2]. Experimental infections with serogroup Icterohaemorrhagiae were successful in young piglets [21,22]. However, in the only documented experimental infection of mature pigs, which was conducted with *Leptospira* serovar Icterohaemorrhagiae in fertile, pregnant sows several decades ago [23], no clinical signs could be initiated. In addition, neither leptospiruria nor leptospiraemia was observed. 

The aim of this preliminary study was to obtain informative results of pathogenicity, transmission, tissue tropism, and immune response after an experimental infection of a restricted number of naïve fertile gilts with a serovar Icterohaemorrhagiae strain and to gain the knowledge required for designing a large-scale trial to clarify the impact of *Leptospira* on pig health, such as reproductive disorders. As described before, high levels of seropositivity were detected against serovar Icterohaemorrhagiae in pigs. For that reason, Icterohaemorrhagiae is considered highly relevant for leptospirosis in swine, but has an unclear pathogenesis because of a lack of scientific studies. Therefore, serovar Icterohaemorrhagiae was chosen for the study. The strain used in the study, route of application, and dosage should be proven to cause a diagnostically manifested infection. The study was performed based on the latest available knowledge and considered animal welfare. The presented as well as future experimental infection trials with *Leptospira* in pigs are important to yield information for scientific purposes, but also for routine practice. 

## 2. Results

### 2.1. Clinical Signs

Within three hours after intravenous inoculation, all three gilts (no. 6, 8, and 9) showed an increase in rectal temperature of >41 °C, which decreased again within the consecutive twelve hours. The general behavior of gilt no. 8 was reduced on D2 and D3 and then again in week 2 on three consecutive days. Gilt no. 6 was the only pig having a slightly icteric sclera on D4. Gilt no. 8 basically discharged small amounts of very concentrated urine during the whole observation period. Apart from those deviations, no clinical signs including vaginal discharge were observed in any of the animals of either group at any time point during the study. No obvious differences in weight gain were recorded between the three groups (data not shown). 

### 2.2. Blood and Urine Chemistry Analysis

Basically, deviations from the reference values were observed mainly in infected gilts (no. 6, 8, and 9). On D4, the urea concentration of gilt no. 6 (infection group) was highly increased with 116 mg/dL (ref.: 15–45 mg/dL). Creatinine (CREA) of all gilts was within the reference range (ref.: <2.26 mg/dL) over the entire trial, albeit there was an increase from 1.3 to 1.9 mg/dL from D4 to D10 in gilt no. 8 (infection group). Total protein (TP) concentration varied in gilts no. 6 and 8 (infection group) but always remained within the reference range. On D4, a low albumin (ALB) value was measured in gilt no. 6 (infection group). The alanine-aminotransferase (ALT) activities of the two control gilts and one contact gilt were within the reference range (<74 U/L) over the entire trial period, but values of the other gilts were already >75 U/L on D0. On D17 ALT of gilt no. 9 (infection group) was elevated (168 U/L) (Figure 1).

On the same day, the aspartate-aminotransferase (AST) value of gilt no. 9 was also highly increased (962 U/L). An increase in AST was generally recorded in two contact animals and all infected gilts after inoculation (Figure 2). 

Gilt no. 6 (infection group) continuously had the highest alkaline phosphatase (ALP) value of all animals. Glutamate-dehydrogenase (GLDH) activity was elevated in gilt no. 6 (infection group) on D7 and D14 (15.43 U/L) (ref.: <8 U/L). No changes were observed for the values of creatinine kinase (CK), sodium (Na), potassium (K), chloride (Cl), and phosphorus (P) during the entire trial.

Five urine samples from different gilts and different time points were missing because the gilts were not urinating during the observation time of two hours. With a few exceptions, most urine pH values were below 7.5 at all time points of sampling, and no differences were observed between the three groups. On D0, the mean pH values were lower than on D28 (6.30 versus 6.78). pH measured by Combur test strips and pH measured with the pH meter achieved roughly the same results.

The urine analytes glucose (GLU), ketones (KET), urobilinogen (UBG), and bilirubin (BIL) measured by the Combur test strips were always unremarkable. Erythrocytes (ERY) and hemoglobin (HB) reactions on the Combur strips were noted on two samples from one control gilt (no. 4) and one infected gilt (no. 6) on D1. Protein (PRO) was detected in the urine of gilts no. 6 and no. 8 (infected group) on D4 and the urine of both control gilts on D28. Values of the control gilts ranged from 1.8 mg/dL to 17.6 mg/dL and values of the infected gilts were between 2.6 mg/dL and 9.7 mg/dL. The same incoherent results could be seen in the creatinine (CREA) analysis. The results of the UPC (urine protein to creatinine ratio) ratio are presented in Figure 3. 

The highest ratios could be demonstrated in infected and contact gilts on D7, indicating higher protein loss through the kidneys in these groups.

### 2.3. Serology

As shown in Figure 4, first agglutination titers beyond the threshold of 1:100 against *Leptospira interrogans* serovar Icterohaemorrhagiae were measured on D4 in two (no. 6 and 8) infected gilts (n = 3) and on D7 in the third infected gilt (no. 9), respectively. While in gilts no. 6 and no. 8 Icterohaemorrhagiae antibodies peaked between D7 and D14 with a maximum titer of 1:1600 and were still present on D28, gilt no. 9 never reached titers exceeding 1:100 and was again seronegative on D24. In gilts no. 6 and no. 8 a four-fold increase in antibody titers was detected between D4 and D7. Noteworthy, none of the contact animals seroconverted. No antibodies were detected in the control animals. Antibodies against serovars Bratislava, Pomona, and Wolffi were also exclusively detected in serum samples from infected gilts. Antibody titers to those serovars were always lower than those against Icterohaemorrhagiae (Appendix A).

### 2.4. Bacteriology and Real-Time PCR Results

No *Leptospira* was detected via microbiological examination of the right ovary, right oviduct, right uterine horn, right kidney, urinary bladder, or liver from any of the eight gilts. All urine samples that were taken throughout the whole trial were negative in bacterial culture for *Leptospira* as well. Equally, real-time PCR for the detection of *Leptospira* spp. DNA was also negative in all investigated samples.

### 2.5. Pathological Findings

In the infected animals, different grades of focal or multifocal, predominantly lymphocytic tubulo-interstitial nephritis were detected in the course of the histologic examination (Figure 5). The inflammatory response was more present in the renal medulla, affecting the collecting ducts. No tubular epithelial alterations were seen, and no spirochaetal organisms could be visualized by Warthin–Starry silver staining. The liver and uterus did not show any relevant histologic alterations.

## 3. Discussion

We were not able to detect *Leptospira* in serum, vaginal swabs, manure, urine, or inner organs of the urogenital tract, and no clinical manifestation of infection was observed, but seroconversion had occurred. The reason for the reduced infectivity and the undetectable strain is possibly related to the limitations of the study. One limitation is the strain used for the intravenous challenge. The passage number of the challenge strain is unknown. In general, high numbers of passages in culture can reduce infectivity. A passage through laboratory rodents, as conducted by some authors to reactivate infectivity and pathogenicity [24,25,26], was not performed. However, it is not entirely clarified if passages can reactivate infectivity and pathogenicity, especially if carried out in different animal species. For example, Feenestad and Borg-Petersen [23] passaged Icterohaemorrhagiae strains prior to challenge. Similar to our study, they could not detect *Leptospira* in the sampled material. Due to the lack of porcine Icterohaemorrhagiae strains, all experimental infection trials have been conducted with human strains, so far [23,26]. This might provide another reason for the low infectivity in pigs. A recently isolated *Leptospira* strain from a pig with a low number of passages would be the best choice for an experimental infection. Limitations due to laboratory passages or differences in infectivity because of the adaptation to a certain host species could be excluded. Unfortunately, to our knowledge, such a strain is not available due to unsuccessful isolation. For a future study, as long as no field isolate is accessible, it is recommendable to conduct a passage of an accessible Icterohaemorrhagiae strain through pigs and to perform pathogenicity testing. 

A further limitation is based on a lack of previous research studies on successful experimental infections of mature pigs with Icterohaemorrhagiae. No information is available on an appropriate route of inoculation and infection dose. We assumed that intravenous application would most probably result in infection since the penetration barrier into the body is pretermitted immediately causing leptospiraemia. The assumption could not be verified in our study. Jacobs et al. [27] infected pregnant sows intravenously and conjunctivally with serovar Pomona (not Icterohaemorrhagiae) and were able to find leptospires in the internal organs and the fetuses. This might be due to the fact that conjunctival application imitates the natural way of infection through skin or mucosa better than intravenous application [2]. Thus, a conjunctival inoculation route or another dermal or mucosal application route might lead to a successful infection with the serovar Icterohaemorrhagiae. This approach of inoculation should be considered for a future study. Regarding the infection dose, saturation was almost reached with the used 10^8^ leptospires/mL, but higher densities up to 10^**9**^ leptospires/mL are possible by cultivation [28]. Furthermore, the infection dose could be raised by increasing the volume. Possibly a higher density and/or volume would have been needed for the used serovar Icterohaemorrhagiae to induce an infection. A consecutive application could be another opportunity for the induction of an infection. As no standardized study design for swine is available, determining the number of leptospires to cause an infection was challenging, and finally, the selected dose was adapted from the infection trial with serovar Pomona by Jacobs et al. [27]. 

The number of animals and therefore sample size is another study limitation. The number is insufficient for statistical measurements, and the results give evidence, but there is no statistical proof. However, it was the intention to perform a preliminary study previously to a large-scale trial to confirm an appropriate strain, route of infection, and infection dose that causes a diagnostically detectable infection. Under the circumstance of limited information for an appropriate study design, a large-scale trial would not have been justifiable, particularly because of animal welfare reasons. 

Besides study limitations, an explanation for the reduced infectivity of Icterohaemorrhagiae in the performed study could be the generally low infectivity of Icterohaemorrhagiae in swine. This circumstance would explain why Icterohaemorrhagiae could not be identified by direct detection methods as the causative agent of reproductive disease in pigs, according to the literature. The existing immunogenicity in swine as shown in our study would explain the widespread antibody detection of Icterohaemorrhagiae in the field [6,7,8]. Feenestad and Borg-Petersen [23], who infected pregnant sows intravenously with several leptospiral isolates, including Icterohaemorrhagiae, reported that sows infected with Pomona developed clinical signs such as abortions and also shed the infectious agent, while sows infected with Icterohaemorrhagiae were completely unsuspicious. In case a low infectivity will be confirmed in future studies, this will be of a high relevance for routine practice. A reconsideration of Icterohaemorrhagiae as a causative agent for reproductive disorders would be required.

Regarding the results of the performed study, inoculation of *Leptospira interrogans* serovar Icterohaemorrhagiae in gilts did not lead to a clinical manifestation of leptospirosis, going in line with the only other Icterohaemorrhaghiae trial in mature pigs (pregnant sows) [23]. Results differ from those for Icterohaemorrhaghiae in piglets with clinical manifestation [21,22]. This might be explained by the immature immune system of neonatal piglets. Due to a high seroprevalence observed in the field [8], we assume that infection with *Leptospira* serovar Icterohaemorrhaghiae happens frequently. Therefore, an early initial infection of piglets is possible. 

In the present study, we used *Leptospira* seronegative gilts. There was a humoral response after Icterohaemorrhaghiae inoculation. Despite the highest measured titer accounting for 1:1.600, it is possible that titers might have been even higher between two time points of sampling. Other experimental studies in pigs with the serovar Pomona confirmed much higher antibody titers of 1:5 × 10^4^ [19] and 1:10^8^ [20]. Low titers are in line with the infection trial conducted by Fennestad and Boerg-Petersen [23], reporting that the highest recorded titers of sows infected with Icterohaemorrhagiae accounted for 1:300. A possible explanation might be the variable immunogenicity of different serovars. Antibody cross-reactions were observed between Icterohaemorrhagiae, Pomona, Bratislava, and Wolffi, respectively, but only in one out of three gilts. Cross-reactivity between serovars is well known and a common finding [29]. 

Multifocal tubulo-interstitial nephritis is a common finding in pigs affected by *Leptospira*, especially by *Leptospira interrogans* serovar Pomona. This condition is characterized by the presence of white spots in the parenchyma of the kidneys [30,31], which was not observed in this study. Michna and Campbell (1969) and Scanziani et al. (1989) [32,33] reported that leptospires are not always detectable in the kidneys of *Leptospira*-infected pigs showing interstitial nephritis, as described in the present case. The histologic detection of tubulo-interstitial nephritis without fibrosis indicates a slight subacute tubulary damage of the affected kidneys. It can be speculated that this was caused by the leptospires, as the lesions were only found in the kidneys of inoculated animals. 

Despite the absence of distinct clinical signs and non-detectable pathogen in the tested samples, elevated rectal temperature briefly after infection, alterations of ALT values in the challenge group, icteric sclera in one animal and tubulo-interstitial nephritis in all animals are indicative of an infection. Furthermore, a four-fold rise in antibody titers as observed in two infected gilts within three days is considered diagnostic [3] and a confirmation of an infection. However, as discussed before, it should be considered that Icterohaemorrhagiae might be immunogenic but not infectious enough to cause a clinical disease manifestation. In that context, it has to be mentioned that, in clinical practice, usually paired serum samples are taken within a larger interval of one week or more. In that case, the infection as presented in our study would have been overlooked. Until present, jaundice in pigs older than three months has not been described in association with infection with the serovar Iceterohaemorrhagiae. 

The lack of further manifested clinical signs is in line with hematologic test results that were mostly within available reference limits. The interpretation of blood tests is very tedious since valid reference values for pigs are not always available [34]. The validity of the latter is debatable due to the lack of evidence for their accuracy since common laboratory testing for companion animals is not standard practice in pig medicine. The results of the Combur test did not always match the results obtained by the biochemistry analyzer, as seen with protein detection in urine. The test strip was also ineffective in the diagnosis of chronic forms of urinary tract diseases by other authors [35]. Noteworthy, the test was not designed for use in pigs. 

Possibly, the absence or reduced infectivity of gilts prevented transmission to the contact animals. The non-contagious status of the gilts during the study is underlined by the negative results of *Leptospira* in excretions (urine and vaginal swabs) and the absence of signs of illness. The infected animals might have shed *Leptospira* at a low level, which was not sufficient for the infection of the contact animals. Even if we had missed the opportunity to demonstrate transmission by direct detection, we would have expected to observe seroconversion in contact animals like in the experimentally infected animals. Nevertheless, since *Leptospira* has a low tenacity in the environment, which is influenced for example by pH and temperature [36], they could have also been shed, but were maybe inactivated too fast to infect contact animals. Furthermore, it might be possible that Icterohaemorrhagiae isolates need an intermediate host (e.g., rats, mice) to stay infectious or to become infectious again for the pig.

Under the conditions of the present study, in fertile gilts, *Leptospira* serovar Icterohaemorrhagiae was not as infectious and pathogenic as expected. In the literature and in routine veterinary practice, the infectivity and pathogenicity of Icterohaemorrhagiae in swine are mostly assumed based on antibody detection. Pivotal questions, including the optimal route and dosage of infection, remain unanswered and should be clarified in future studies. The limitation of the lacking strain isolated from a pig with a low number of passages could be eliminated by research efforts focusing on the isolation and typing of field strains from pigs. Thus, more future research on *Leptospira* is necessary to get a deeper knowledge of the complex disease leptospirosis.

## 4. Materials and Methods

Eight conventionally raised six-month-old fertile non-pregnant gilts (German Landrace x Large White), seronegative (MAT titers ≤ 1:50) for the *Leptospira* serovars Icterohaemorrhagiae, Bratislava, Canicola, Grippotyphosa, Pomona, Wolffi, Tarassovi, and Hardjo, were brought into the animal biosafety level 2 facilities of the University Clinic for Swine, Vetmeduni Vienna. All animals were housed in isolation units. They were divided into three groups by simple randomization: control group, no. 4 and 7 (n = 2), infection group, no. 6, 8, and 9 (n = 3), and contact group, no. 5, 13, and 15 (n = 3). The control group was housed separately from the two other groups. The contact group had direct contact with the infected gilts. According to Austrian law, all gilts were fed ad libitum with a commercial diet without the addition of antibiotics and had permanent access to fresh water and enrichment material. Starting seven days prior to inoculation (D-7), a daily clinical examination, including measurement of rectal temperature, general behavior, feed intake, evaluation of the mucous membranes of the mouth and eyes, as well as the presence of vaginal discharge, was performed. Gilts were weighted at D0 and D28. At D0, the infection group and the contact group were separated for three hours to avoid indirect infection with the inoculum. The challenge group was infected intravenously with *Leptospira interrogans* serovar Icterohaemorrhagiae by application of the inoculum (10^8^ leptospires/mL liquid culture medium per animal, total volume 10 mL) in the left lateral ear vein using a venous catheter. 

Icterohaemorrhagiae RGA strain (isolate from a diseased human from Europa) was obtained by the Institute for Veterinary Disease Control, Modeling, Austrian Agency for Health and Food Safety (AGES) at an unknown passage level from the Academic Medical Centre (Leptospirosis Reference Centre, Amsterdam, Netherlands). No Icterohaemorrhagiae strain from swine and/or with a known passage level was accessible, for that reason the mentioned strain was used. Five, four, and three days before injection of the inoculum, the strain was transferred into tubes with 10 mL fresh culture medium (Ellinghausen McCullough, Johnson, and Harris; EMJH). On D0, optical density was measured, and the viability of the strain was examined by dark field microscopy. The cultures of day four showed the highest density and best viability. These cultures were merged into one tube, the density (10^8^ leptospires/mL) was measured with a Helber Counting Chamber as described elsewhere [28], and the tube was transported (half an hour by car) to the University Clinic for Swine, Vienna, where the inoculum was injected within two hours. 

Blood samples from the jugular vein, urine samples, and vaginal dry swabs (sterile dry swabs, Copan Italia S.p.A., Brescia, Italy) were collected from each individual on D0, D2, D4, D7, D10, D14, D17, D21, D24, and D28, always at the same time (9 a.m.). For serum collection (Primavette® V Serum 9 mL KABE LABORTECHNIK GmbH, Nümbrecht- Elsenooth, Germany), blood was centrifuged (10.000× *g*, 10 min), and serum was stored at −20 °C until further testing. CREA, CK, urea, symmetrical dimethylarginine (SDMA), AST, ALT, ALP, GLDH, TP, ALB, bile acids, total bilirubin, lipase, Na, K, Cl, and P were measured in lithium heparin plasma samples (Primavette^®^ V Li.-Heparin 10 mL KABE LABORTECHNIK GmbH, Nümbrecht- Elsenooth, Germany). Analyses were performed on a fully selective biochemistry analyzer (Hitachi 501c™, Roche Diagnostics, Vienna, Austria) by standardized methods applied according to the manufacturer’s recommendations. All used reference intervals are from the Central Laboratory of the Vetmeduni Vienna, which is an accredited laboratory (ISO 9001: 2015). Serum biochemistry was analyzed using a Cobas 6000 c501 automated biochemistry analyzer (Roche Diagnostics, Austria) and commercial kits (Roche), except for BA (Randox) and SDMA (Eurolyser). Urine samples were immediately analyzed by measuring the pH using a pH meter (Testo 206 pH measuring instrument, TESTO, Germany) and urine parameters (pH, PRO, GLU, KET, UBG, BIL, ERY, and HB) using the Combur test^®^ (Roche, Switzerland). One microliter of urine was transferred into Eppendorf tubes for immediate chemical analysis of urinary creatinine and protein, from which the UPC ratio was calculated. The assays were performed on the analyzer described above. Further urine processing took place within two hours after collection. 1.5 mL of urine were transferred into Eppendorf tubes, centrifuged (13.000× *g*) for 20 minutes, the supernatant was decanted, and the pellet was frozen at −20 °C for real-time PCR testing at a later stage. Another 100 µL of urine was transferred into 10 mL of EMJH-STAFF culture medium and placed in a heating chamber at 29 °C to provide optimized conditions for leptospiral growth. The EMJH-STAFF medium was prepared as follows: Liquid EMJH medium (DifcoTM *Leptospira* Medium Base EMJH and *Leptospira* Enrichment EMJH, Becton Dickinson) was generated according to the manufacturer´s instructions, and the selective agents STAFF (40 μg sulfamethoxazole, 20 μg trimethoprim, 5 μg amphotericin, 200 μg fosfomycin, and 100 μg 5-fluorouracil in 1 mL EMJH medium) were added. Vaginal swabs were stored at −20 °C until further analysis by real-time PCR. Liquid manure was collected from both rooms once a week (n = 4) and tested via real-time PCR for *Leptospira* DNA. 

Gilts of all three groups were euthanized for necropsy on D28 by intravenous injection with ketamine hydrochloride (Narketan^®^10 ad us.vet., Vetoquinol AG) and azaperone (Stresnil^®^ ad us. vet., Elanco Tiergesundheit AG) followed by an intracardial injection of T61^®^ (MSD Tiergesundheit). Tissue samples (liver, right kidney, right uterine horn) obtained during necropsy were fixed in 7.5% neutral buffered formalin and processed into paraffin wax. Sections were cut at 3–4 µm and stained with hematoxylin and eosin (H&E) and Warthin–Starry silver stain for histologic examination. Tissue samples from the liver, kidneys, urinary bladder, ovaries, oviducts, and both uterine horns were put into 10 mL of EMJH-STAFF medium (pieces of approximately 1 cm^3^) for leptospiral culture and into Eppendorf tubes, which were stored at −20 °C until further testing by real-time PCR. 

Serology, real-time PCR analysis, and cultivation were performed in the diagnostic laboratory of the Institute for Veterinary Disease Control, Modeling, Austrian Agency for Health and Food Safety (AGES). All collected sera were tested by the same person for antibodies against the serovars Icterohaemorrhagiae, Bratislava, Canicola, Grippotyphosa, Pomona, Wolffi, Tarassovi, and Hardjo with the MAT as described elsewhere [28]. A serial dilution from 1:50 to 1:6.400 was performed. Furthermore, the sera, urine pellets, vaginal swabs, tissue samples, and manure were tested for *Leptospira* spp. by real-time PCR targeting the leptospiral outer membrane lipoprotein gene lipL32 [37] following sample type-specific pre-treatment and nucleic acid extraction; no pre-treatment was performed for serum samples. Urine pellets were dissolved in 140 µL PBS. Vaginal swabs were reconstituted in 300 µL PBS and vortexed vigorously. Tissue (about double the size of a rice grain—approx. 20–25 mg) was homogenized in lysis buffer (180 µL ATL-buffer, 200 µL PBS, 20 µL proteinase K) by beating with a 5 mm steel bead in a TissueLyzer II (Qiagen, Germany) for 4 min at 20 Hz. One hundred microliters of serum or supernatant from urine pellets or vaginal swabs or 200 µL of tissue homogenate was used for nucleic acid extraction with a magnetic bead-based extraction kit (BioExtract^®^ SuperBall^®^, BioSellal, France) on the KingFisher™ Flex Purification System (Thermo Fisher Scientific, Austria). Each tissue sample for leptospiral culture was transferred to a homogenizer (Homogenisator E 3030, AES Laboratoire, France) within six hours of sampling, together with phosphate-buffered NaCl, and homogenized for ten minutes. Further, 100 μL of the homogenate was transferred to 10 mL EMJH-STAFF medium and incubated at 29 °C. Urine and organ cultures were evaluated for leptospiral growth by dark field microscopy (200×) once a week for twelve weeks. Laboratory diagnostic investigations were performed blinded. The investigators had no information about the animal identity. 

## Figures and Tables

**Figure 1 pathogens-12-00135-f001:**
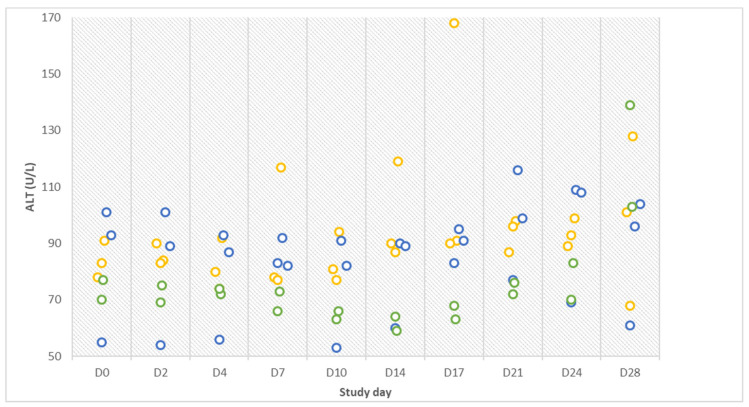
Alanine-aminotransferase (ALT) results of infected (orange), contact (blue), and control (green) animals on ten different study days over the entire course of the study.

**Figure 2 pathogens-12-00135-f002:**
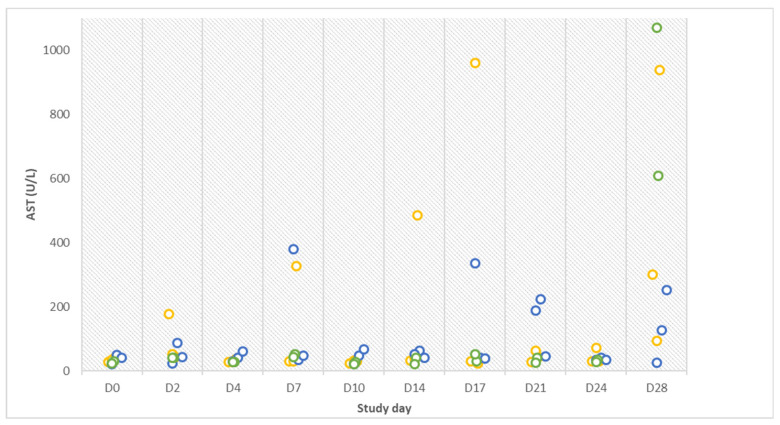
Aspartate-aminotransferase (AST) results in infected (orange), contact (blue), and control (green) animals on ten different study days over the entire course of the study.

**Figure 3 pathogens-12-00135-f003:**
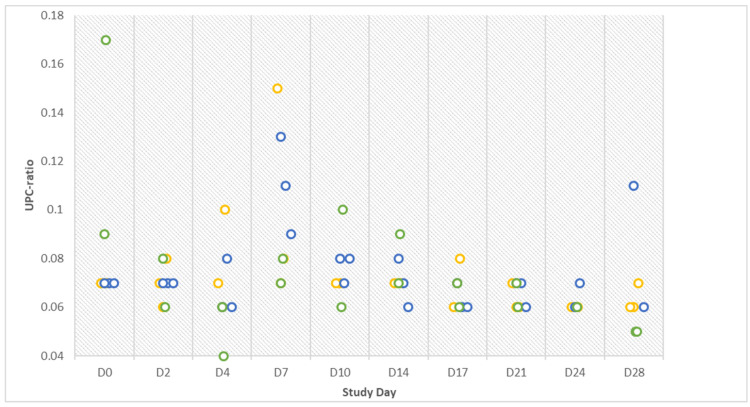
Urine protein to creatinine ratio (UPC) results of infected (orange), contact (blue), and control (green) animals on ten different study days over the entire course of the study.

**Figure 4 pathogens-12-00135-f004:**
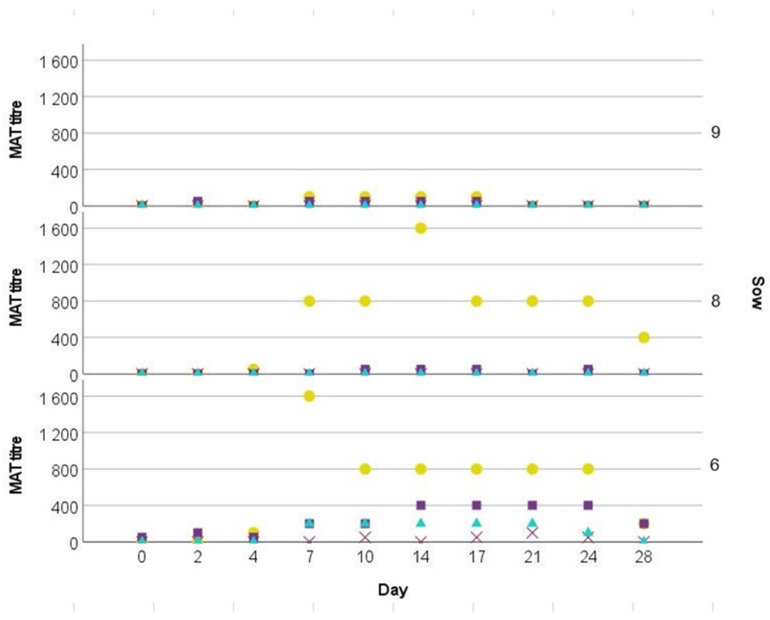
Microscopic agglutination test (MAT) titres of infected gilts (no. 6, 8, and 9) of four *Leptospira* serovars on ten study days. Yellow dot: Icterohaemorrhagiae, purple quare: Bratislava, purple cross: Pomona, turquoise triangle: Wolffi.

**Figure 5 pathogens-12-00135-f005:**
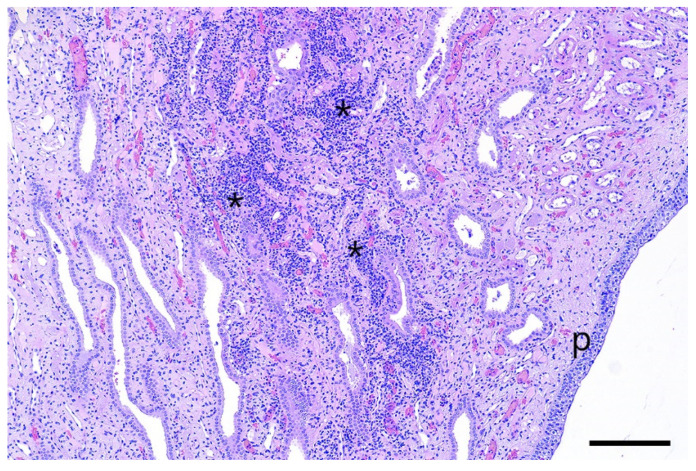
Severe tubulo-interstitial nephritis (asterisks) near to the renal pelvis (p) of an infected gilt. No tubular alterations are seen. Microphoto, H&E, bar= 200 µm.

## Data Availability

Not applicable.

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
