# Peer review of "Leptospira interrogans Serovar Icterohaemorrhagiae Failed to Establish Distinct Infection in Naïve Gilts: Lessons Learned from a Preliminary Experimental Challenge"

_pathogens, 2023, doi:10.3390/pathogens12010135_

Round 1

Reviewer 1 Report (Previous Reviewer 1)

Quality of the manuscript was improved. I suggest the change of grafics in the presentation of the results

Since some of the comments have not been taken into account I suggest it would be good to consider them in the future study with the aim to induce diagnostically detectable infection :

Use of pathogenic isolate of any origin with higher density

Change the route of application

Use of commercially bred animals is not the best option

Quality of the manuscript was improved. I suggest the change of grafics in the presentation of the results

Since some of the comments have not been taken into account I suggest it would be good to consider them in the future study with the aim to induce diagnostically detectable infection :

Use of pathogenic isolate of any origin with higher density

Change the route of application

Use of commercially bred animals is not the best option

Author Response

Reviewer 2 Report (New Reviewer)

Generally: the proposed manuscript is very well written.

            In details:  

1) Abstract is concise and relevant about the importance of the studied problem and the main purpose is clear. The mentioned Keywords are appropriate for the studied issue.

2) The Introduction gives enough information about the studied problem. The authors mentioned: “The aim of this preliminary study was to obtain informative results of pathogenicity, transmission, tissue tropism and immune response after an experimental infection of a restricted number of naïve fertile gilts with a serovar Icterohaemorrhagiae strain and to gain required knowledge for designing a large-scale trial to clarify the impact of Leptospira on pig health like reproductive disorders“. The mentioned references are cited correctly.

3) About the section Results:

The Study Design is well structured and relevant for achievement of the aim of the study. In sub-sections 2.1. Clinical signs, 2.2. Blood and urine chemistry analysis, 2.3. Serology, 2.4. Bacteriology and real-time PCR results, and 2.5. Pathological findings  all received information is described precisely. The results are well visualized by five figures.

4) In the section Discussion the authors mentioned the limitations of the study as follows: 1) “The reason for the reduced infectivity and the undetectable strain is possibly related to the limitations of the study. One limitation is the used strain for the intravenous challenge.”; 2) “Due to the lack of porcine Icterohaemorrhagiae strains all experimental infections trials were conducted with human strains, so far [23, 26]. This might provide another reason for the low infectivity in pigs. A recently isolated Leptospira strain from a pig with a low number of passages would be the best choice for an experimental infection. Limitations due to laboratory passages or differences in infectivity be cause of the adaptation to a certain host species could be excluded. Unfortunately, to our knowledge such a strain is not available due to unsuccessful isolation. For a future study, as long as no field isolate is accessible, it is recommendable to conduct a passage of an accessible Icterohaemorrhagiae strain through pigs and to perform pathogenicity testing.“; 3) “A further limitation is based on a lack of previous research studies on successful experimental infections of mature pigs with Icterohaemorrhagiae. No information is available on an appropriate route of inoculation and infection dose. We assumed that intravenous application would most probably result in infection, since penetration barrier into the body is pretermitted immediately causing leptospiraemia. The assumption could not be verified in our study.”; 4) “The number of animals and therefore sample size is another study limitation. The number is insufficient for statistical measurements and the results give an evidence but there is no statistical proof. However, it was the intention to perform a preliminary study previously to a large-scale trial to confirm an appropriate strain, route of infection and infection dose that cause a diagnostically detectable infection. Under the circumstance of limited information for an appropriate study design, a large-scale trial would not have been justifiable particularly because of animal welfare reasons. I am fully agreed with these statements. The explained limitations do not minimalize the valuable considerations of this interesting study. The conclusions are relevant and based on the results of the study. The authors emphasized: “Under the conditions of the present study, in fertile gilts Leptospira serovar Icterohaemorrhagiae was not as infectious and pathogenic as expected. In literature and in routine veterinary practice, infectivity and pathogenicity of Icterohaemorrhagiae in swine is mostly assumed based on antibody detection. Pivotal questions including the optimal route and dosage of infection remain unanswered and should be clarified in future studies. The limitation of the lacking strain isolated from a pig with a low number of passages could be eliminated by research efforts focusing on the isolation and typing of field strains from pigs. Thus, more future research on Leptospira is necessary to get a deeper knowledge of the complex disease leptospirosis.      

5) In the section Methods, the authors precisely described all used methods. The authors mentioned: “The number of animals and therefore sample size is another study limitation. The number is insufficient for statistical measurements and the results give an evidence but there is no statistical proof.

6) In the list of References are included 37 sources of information (13 of them from last five years – 35.14%). The articles are listed in order of mentioning in the text and are cited correctly.

            At the end of my notes I conclude that this study is on high level and deserves to be published. It will be of strong interest to the readers of the journal.  

My final recommendation is to accept the manuscript in present form.  

Author Response

This manuscript is a resubmission of an earlier submission. The following is a list of the peer review reports and author responses from that submission.

Round 1

Reviewer 2 Report

ABSTRACT, TITLE AND REFERENCES

The paper describes a clinical infection in naïve gilts with Leptospira Icterohaemorrhagiae with the aim to gain required knowledge for designing a large-scale trial.

The abstract explains the goal of the research and some of the main findings.

The title recalls the findings of the study and the references are partially recent; a number of references includes some key studies on experimental Leptospira infection.

INTRODUCTION

The introduction briefly summarizes the current general knowledge about the epidemiology of leptospirosis: it is supported by a number of references giving a general overview on what is known about the topic.

The authors clearly state there are some gaps of knowledge in epidemiology and pathogenesis of leptospirosis and that these gaps could be filled with experimental trials.

Referring to the serovar Icterohaemorrhagiae the authors cite its relatively high frequency in serological test but not associated with clinical signs. They start from this point to support their experimental infection with Leptospira Icterohaemorrhagiae in gilts having the aim to obtain results that can explain different aspects of the infection.

MATERIALS AND METHODS

The methods described are well-known, valid, reliable and reproducible.

The authors should cite the bibliography where they found the reference values they used to assess the hematologic test (see also discussion: lines 244-245).

RESULTS

The data are presented appropriately: figures are clear with titles labelled correctly and clearly. The text summarizes briefly the contents of the tables but it is not repetitive.

The results confirm the role of serovar Icterohaemorrhagiae when infecting pigs: no clinical signs, limited and non-visible lesions, seroconversion. Leptospira was not detected and this probably has to be looked for in the characters of the infection itself and in the method of detection as well.

DISCUSSION AND CONCLUSIONS

The authors start the discussion admitting that the lack of detection of Leptospira might be due to a number of reasons one of them appears to be a flaw in the experimental design.

The challenge strain is not well known in its characters and, in particular, the passage number is unknown. So, a lack of virulence (due to a high number of passages) might be an explanation of the low infectivity. They also, as they report, did not pass the strain through laboratory animals.

The challenge dose was adapted from a previous trial conducted with Leptospira Pomona but this does not appear suitable, considering that serovar Pomona has a different behaviour, as a host in pigs, than serovar Icterohaemorrhagiae. Moreover, the authors do not explain how they determined the used challenge dose (they use a lower dose than in trials reported in the reference 28).

The validity of hematologic tests, as the authors say, is debatable because of the lack of certain reference values: may be the results are an aid for a general overview (e.g. in a comparison between challenged and non-challenged animals) but they may not be considered meaningful.

Given that Leptospira is not easy to isolate, it must be said that the choice of a liquid medium is less suitable, for isolation, than a semi-solid medium; moreover, the initial concentration of the suspension in the medium (100 microliters of urine or homogenate in 10 mL of medium) looks too low to have a chance of isolation.

FINAL COMMENT

The study design looks appropriate to answer the aim but there are some limitations that makes the article unsuitable for publication. Nevertheless, as the authors state, the goal is also to collect information for subsequent trials – so these limitations are opportunities to inform a future research.

Round 2

Reviewer 2 Report

The corrections look like insufficient, compared with the first version of the manuscript which still remains unsuitable for pubblication. Anyway, as previously stated, the collected data might be useful for a future research.